# Variability in the Chemical Composition of *Myrcia sylvatica* (G. Mey) DC. Essential Oils Growing in the Brazilian Amazon

**DOI:** 10.3390/molecules27248975

**Published:** 2022-12-16

**Authors:** Jamile Silva da Costa, Jofre Jacob da Silva Freitas, William N. Setzer, Joyce Kelly R. da Silva, José Guilherme S. Maia, Pablo Luis B. Figueiredo

**Affiliations:** 1Programa de Pós-Graduação em Ciências Farmacêuticas, Universidade Federal do Pará, Belém 66075-110, Brazil; 2Laboratório de Química dos Produtos Naturais, Universidade do Estado do Pará, Belém 66087-662, Brazil; 3Laboratório de Morfofisiologia Aplicada a Saúde, Departamento de Morfologia e Ciências Fisiológicas, Universidade do Estado do Pará, Belém 66087-662, Brazil; 4Aromatic Plant Research Center, 230 N 1200 E, Suite 100, Lehi, UT 84043, USA; 5Programa de Pós-Graduação em Biotecnologia, Universidade Federal do Pará, Belém 66075-900, Brazil; 6Programa de Pós-Graduação em Química, Universidade Federal do Maranhão, São Luís 64080-040, Brazil; 7Departamento de Ciências Naturais, Centro de Ciência Sociais e Educação, Universidade do Estado do Pará, Belém 66050-540, Brazil

**Keywords:** chemical variability, sesquiterpenes, multivariate analyses

## Abstract

*Myrcia sylvatica* (G. Mey) DC. is known as “insulin plant” because local communities use the infusions of various organs empirically to treat diabetes. The leaves of seven specimens of *Myrcia sylvatica* (Msy-01 to Msy-07) were collected in the Brazilian Amazon. Furthermore, the essential oils were extracted by hydrodistillation and analyzed by gas chromatography coupled to mass spectrometry, and their chemical compositions were submitted to multivariate analysis (Principal Component Analysis and Hierarchical Cluster Analysis). The multivariate analysis displayed the formation of four chemical profiles (chemotypes), described for the first time as follows: chemotype I (specimen Msy-01) was characterized by germacrene B (24.5%), γ-elemene (12.5%), and β-caryophyllene (10.0%); chemotype II (specimens Msy-03, -06 and -07) by spathulenol (11.1–16.0%), germacrene B (7.8–20.7%), and γ-elemene (2.9–7.6%); chemotype III (Msy-04 and -05) by spathulenol (9.8–10.1%), β-caryophyllene (2.5–10.1%), and δ-cadinene (4.8-5.6%); and chemotype IV, (Msy-02) by spathulenol (13.4%), caryophyllene oxide (15.0%), and α-cadinol (8.9%). There is a chemical variability in the essential oils of *Myrcia sylvatica* occurring in the Amazon region.

## 1. Introduction

The *Myrcia* genus has about 800 species distributed from Central to Tropical America [1]. In addition, it is considered one of the most taxonomically and morphologically complex homogeneous genera of the Myrtaceae family [2], including in the Myrtales order, Rosideas clade, and Malvideas sub-clade [3]. The Amazon rainforest, despite comprising a low diversity of *Myrcia* spp., was an important region in the biogeographic history of this genus because evidence indicates that it participated in the diversification of ancestral lineages [4].

*Myrcia* species have great ecological relevance, as their fruits are a food source for ants, birds, and mammals, and their flowers are attractive to pollinators, such as bees. These ecological relationships are responsible for promoting the conservation of the diversity of this genus [5]. In addition, *Myrcia* species have economic, nutritional [6], and medicinal importance [7].

*Myrcia* species can be recognized in the field by the sweet aroma emanating from the leaves, flowers, and fruits, in addition to generally appearing as shrubs with leaves elliptical-co-lanceolate; apex long-acuminate to caudate; inflorescences in panicles; and flowers with deltoid sepals and petals white—rarely yellow—connective with glands of blackish color and stigma hairy at the base [8].

Several species of the *Myrcia* genus are popularly known as “pedra-ume-caá” or “pedra-hume-caá”, among them *Myrcia punicifolia* (Kunth) DC., *M. speciosa* (Amsh.) Mc Vaugh, *M. amazonica* DC., *M. citrifolia* (Aubl.) Urb., *M. guianensis* (Aubl.) DC., *M. multiflora* (Lam.) DC., *M. salicifolia* DC., *M. sylvatica* (G. Mey) DC., and *M. uniflora* DC. These species are also as known “insulin plant” because local communities use the infusions of various organs of these plants empirically to treat diabetes [9].

*Myrcia sylvatica* (G. Mey) DC. is also known as “kumate-folha-miúda” or “murtinha”. It is native and non-endemic to Brazil, widely distributed in South America, where it is found from Guyana to Brazil [10]. However, in Brazil, its occurrence is restricted to the phytogeographic domains of the Amazon, Caatinga, and Cerrado [11].

The *M. sylvatica* essential oil have shown great chemical variability due to intraspecific or seasonal variations [9,12], in addition to antioxidant, anesthetic potential [13] and bactericidal properties [14].

Therefore, in view of the biological potential presented by *Myrcia sylvatica*, the objective of this work was to investigate the chemical variability of the essential oils of leaves that occur in the Amazon of Pará.

## 2. Results and Discussion

### 2.1. Yield and Chemical Composition of the Essential Oils

The seven *Myrcia sylvatica* wild specimens evaluated in this work showed chemical variability of their essential oils. The oil yield ranged from 0.3 to 0.9%, as shown in Table 1. The quantification and identification of 112 constituents in the analyzed oils represent an average of 81.1% of the total oil content.

Sesquiterpene hydrocarbons (12.5–71.8%) and oxygenated sesquiterpenoids (17.4–71.5%) were predominant in the essential oils. The main compounds (>5%) identified in the oils were the sesquiterpenes with germacrane (germacrene B, 0.3-24.5%; γ-elemene, 0.3–12.5%), aromadendrane (spathulenol, 2.9–16.0%; globulol, 0.0–7.4%; and viridiflorol, 0.8–5.3%), and caryophyllane skeletons (caryophyllene oxide, 0.1–15.0%; and β-caryophyllene, 1.8–10.1%), followed by sesquiterpenes with cadinane skeletons (α-cadinol, 1.7–8.9%; muurola-4,10(14)-dien-1-β-ol, 0.0–5.8%; δ-cadinene, 1.1–5.6%; and *epi*-α-cadinol, 0.0–5.1%), as shown in Figure 1.

The seasonal and circadian study of essential oil from leaves and fruits of *M. sylvatica* collected in the municipality of Santarém, state of Pará, indicated that the yield varied from 0.9 to 1.7% [12], values higher than this work. In contrast, the yield of leaf essential oil from this species collected in Carolina, state of Maranhão, was 0.5% [17], the same content presented by the specimen Msy-07.

In the Myrtaceae species essential oils, the predominance of hydrocarbon and oxygenated sesquiterpenes has been evidenced, some of them with biological properties [18,19]. The presence of the sesquiterpene hydrocarbon β-caryophyllene (45.0%) as the major constituent was identified in a *M. sylvatica* sample collected in Maranhão [17]. Other compounds were also reported as the main compound in oils from Tocantins, among them the oxygenated sesquiterpenes spathulenol (13.8–40.2%) and caryophyllene oxide (5.0–16.6%) [10]. Germacrene B (6.7%) and γ-elemene (10.5%) were identified as the highest content in *M. splendens* [20].

### 2.2. Chemical Variability in the Specimens

The Hierarchical Cluster Analysis (HCA, Figure 2) and the Principal Components Analysis (PCA, Figure 3), carried out with the compounds in the highest abundance (> 4.0%) in the essential oils of *M. sylvatica*, displayed the formation of four groups (chemotypes).

The Principal Components Analysis elucidated 81.5% of the data variability. PC1 explained 42.3% and showed positive correlations with the constituents spathulenol (r = 0.22), caryophyllene oxide (r = 0.33), viridiflorol (r = 0.29), muurola-4,10(14)-dien-1β-ol (r = 0.36), *epi*-α-murrolol (r = 0.34), and α-cadinol (r = 0.36). The second component explained 22.3% and presented a positive correlation with the compounds β-caryophyllene (r = 0.07), germacrene D (r = 0.43), δ-cadinene (r = 0.48), caryophyllene oxide (r = 0.05), viridiflorol (r = 0.08), α-cadinol (r = 0.19), and *epi*-α-cadinol (r = 0.47). The third component, PC3, explained 17.0% of the data and explained a positive correlation with the variables β-caryophyllene (r = 0.18), γ-elemene (r = 0.33), germacrene D (r = 0.05), bicyclogermacrene (r = 0.25), germacrene B (r = 0.23), caryophyllene oxide (r = 0.33), viridiflorol (r = 0.23), muurola-4,10(14)-dien-1β-ol (r = 0.25), *epi*-α-cadinol (r = 0.13), and α-cadinol (r = 0.16).

From this, the oil samples were classified into four chemotypes (chromatogram displayed in Figure A1). Group I (specimen Msy-01) was characterized by germacrene B (24.5%), γ-elemene (12.5%), and β-caryophyllene (10.0%). Group II (Msy-03, -06 and -07 specimens) was characterized by the contents of spathulenol (11.1–16.0%), germacrene B (7.8–20.7%), and γ -elemene (2.9–7.6%). Group III (Msy-04 and -05) showed spathulenol (9.8–10.1%), β-caryophyllene (2.5–10.1%), and δ-cadinene (4.8–5, 6%). Group IV (Msy-02) was characterized by spathulenol (13.4%), caryophyllene oxide (15.0%), and α-cadinol (8.9%).

Three chemical profiles of *M. sylvatica* samples collected in Tocantins were reported, the first one exhibiting selin-11-en-4α-ol (24.7%), caryophyllene oxide (16.6%), and spathulenol (13.8%) as the main constituents. The second was characterized by *cis*-calamenene (30.1%), spathulenol (18.7%), and α-calacorene (11.5%), and the third by spathulenol (40.2%) and β-bisabolene (14.7%) [10]. The oxygenated sesquiterpene spathulenol was present in all samples of this work. Saccol et al., analyzing the chemical composition and the anesthetic and antioxidant effects of *M. sylvatica* essential oil, identified β-selinene (9.96%), cadalene (9.36%), α-calacorene (9.17%), and (*Z*)-calamene (8.17%) as major compounds [13], which is different from the chemical profiles of this study.

Furthermore, the seasonal and circadian study of a specimen of *M. sylvatica* from Santarém, Pará, revealed the influence of climatic factors on the chemical composition of the oils of this species, whose main constituents during the collection period were β-selinene (6.2–10.5%), 1-*epi*-cubenol (5.9–9.8%), cadalene (1.5–6.5%), mustakone (2.7–6.2%), α-calacorene (1.5–6.2%), δ-cadinene (0.7–6.0%), cubenol (2.4–4.6%), *trans*-calamenene (3.5–6.5%), and caryophyllene oxide (2.5–4.0%) [12]. All these compounds were also present in the oils of the studied *M. sylvatica* specimens.

In another study, carried out by Silva et al. [14], the chemical composition of fresh and dried leaves of *M. sylvatica*, also collected in Santarém, exhibited the compounds 1-*epi*-cubenol (6.9–9.9%), *ar*-curcumene (1.9–7.6%), cadalene (5.8–7.2%), β-selinene (6.0–7.0%), β-calacorene (5.4–5.5%), *cis*-calamenene (4.8–5.2%), *ar*-turmerol (0.0–4.9%), muskatone (3.4–4.4%), δ-cadinene (4.2%), and cubenol (4.2%). Only the constituents *ar*-curcumene and *cis*-calamenene were not identified in the samples of this work.

Another specimen collected in Bujaru, Pará state, was rich in (*Z*)-*trans*-α-bergamotene (24.6%), followed by α-sinensal (13.4%), (*Z*)-α-bisabolene (8.3%), *trans*-α-bisabolene (7.1%), and *trans*-β-bisabolene (5.1%). These constituents were not identified in the collected specimens [21]. The oil extracted from a specimen collected in the state of Maranhão showed β-caryophyllene (45.9%), hydroxy-(*Z*)-caryophyllene (10.2%), β-selenene (5.9%), and seline-3,11-diene (5.4%) in higher content [17]. The sesquiterpenes β-caryophyllene and β-selenene were also identified in the oils of the *M. sylvatica* described in this work.

Essential oils from Myrtaceae species have shown chemical variability, which may be influenced by seasonality, collection site, extraction method, genetics, and plant part [18,22,23]. This variability affects their biological properties and applications; for example, the existence of four *Eugenia uniflora* chemotypes was reported, and the samples presented different biological potentials related to their chemical profiles [24].

Therefore, among the collected samples, all chemical profiles were described for the first time: Profile I (germacrene B, γ-elemene, and β-caryophyllene), Profile II (spathulenol, germacrene B, and γ-elemene), Profile III (spathulenol, β-caryophyllene, and δ-cadinene), and Profile IV (spathulenol, caryophyllene oxide, and α-cadinol). Thus, added to the eight chemotypes described in the literature, it is possible that there are at least twelve *Myrcia sylvatica* chemotypes. The occurrence of different chemical profiles can be attributed to the genetic variability of this species [9].

## 3. Materials and Methods

### 3.1. Plant Material

The leaves of the seven *Myrcia sylvatica* wild-growing specimens were collected on Caratateua Island, Belém, Pará state, Brazil, during the rainy season. The collection site, herbarium voucher number, and geographic coordinates are listed in Table 2. After identification, the plant specimens were deposited in the Herbarium of Museu Paraense Emílio Goeldi (MG) in the city of Belém, Brazil. The leaves were dried for three days at room temperature, ground, and then submitted to essential oil hydrodistillation in duplicate using a Clevenger-type apparatus. The oils obtained were dried over anhydrous sodium sulfate, and total oil yields were expressed as mL/100 g of the dried material [25,26]. The specimens were collected in agreement with the Brazilian laws concerning the protection of biodiversity (SISGEN A78F864).

### 3.2. Analysis of Essential Oil Composition

The oil composition analysis was performed by GC-MS, using a Shimadzu instrument Model QP-2010 ultra (Shimadzu, Tokyo, Japan) equipped with a Rtx-5MS (30 m × 0.25 mm; 0.25 μm film thickness) fused silica capillary column (Restek, Bellefonte, PA, USA). Helium was used as carrier gas, adjusted to 1.0 mL/min at 57.5 KPa; split injection (split ratio 1:20) of 1 μL of *n*-hexane solution (oil 5 μL: *n*-hexane 500 μL); injector and interface temperature were 250 °C; oven programmed temperature was 60 to 240 °C (3 °C/min), followed by an isotherm of 10 min. EIMS (electron impact mass spectrometry): electron energy, 70 eV; ion source temperature was 200 °C. The mass spectra were obtained by automatically scanning every 0.3 s, with mass fragments in the range of 35–400 *m*/*z*. The compounds present in the samples were identified by comparison of their mass spectrum and retention index, calculated for all volatile components using a linear equation by Van den Dool and Kratz [27], with the data present in the commercial libraries FFNSC-2 [16] and Adams [15]. The retention index was calculated using *n*-alkane standard solutions (C8–C40, Sigma-Aldrich, St. Louis, MO, USA) under the same chromatographic conditions. The GC-FID analysis was carried out on a Shimadzu QP-2010 instrument, equipped with an FID detector, in the same conditions, except that hydrogen was used as the carrier gas. The percentage composition of the oil samples was computed from the GC-FID peak areas. The analyses were carried out in triplicate.

### 3.3. Multivariate Statistical Analyses

The data matrix was standardized for the multivariate analysis by subtracting the mean and then dividing it by the standard deviation. The hierarchical grouping analysis (HCA), considering the Euclidean distance and complete linkage, was used to verify the similarity of the oil samples based on the distribution of the constituents selected. The principal component analysis (PCA) was applied to verify the interrelation among the oils’ components (>4%) (OriginPro trial version, OriginLab Corporation, Northampton, MA, USA).

## 4. Conclusions

The intraspecific chemical variability among the *Myrcia sylvatica* specimens studied was evidenced by the occurrence of four chemotypes, described here for the first time, with a predominance of the sesquiterpenes class in all samples. In addition to the chemotypes already described in the literature (8 chemotypes), it is possible that at least 10 *Myrcia sylvatica* chemotypes occur. Considering the potential of *M. sylvatica*, the knowledge of this variability can contribute to chemotaxonomy, economical use, and future studies that evaluate the biological properties of this species.

## Figures and Tables

**Figure 1 molecules-27-08975-f001:**
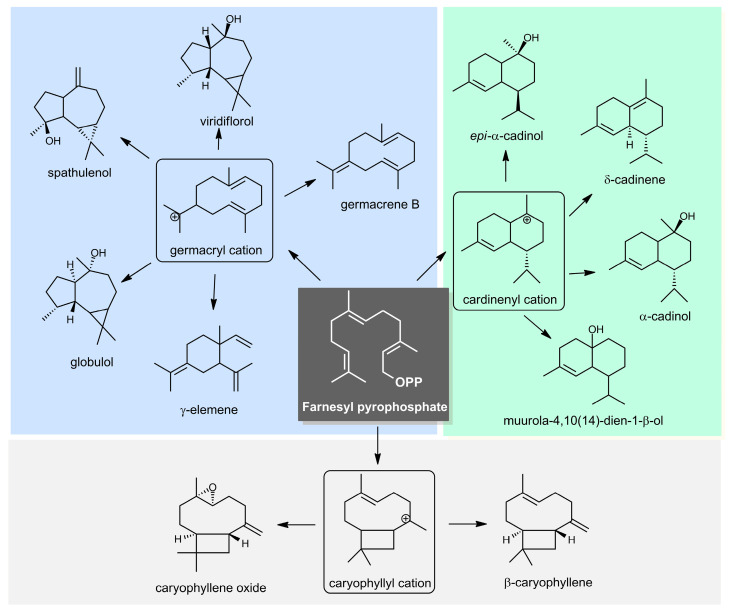
Biosynthetic pathway of the main constituents from *Myrcia sylvatica* essential oil.

**Figure 2 molecules-27-08975-f002:**
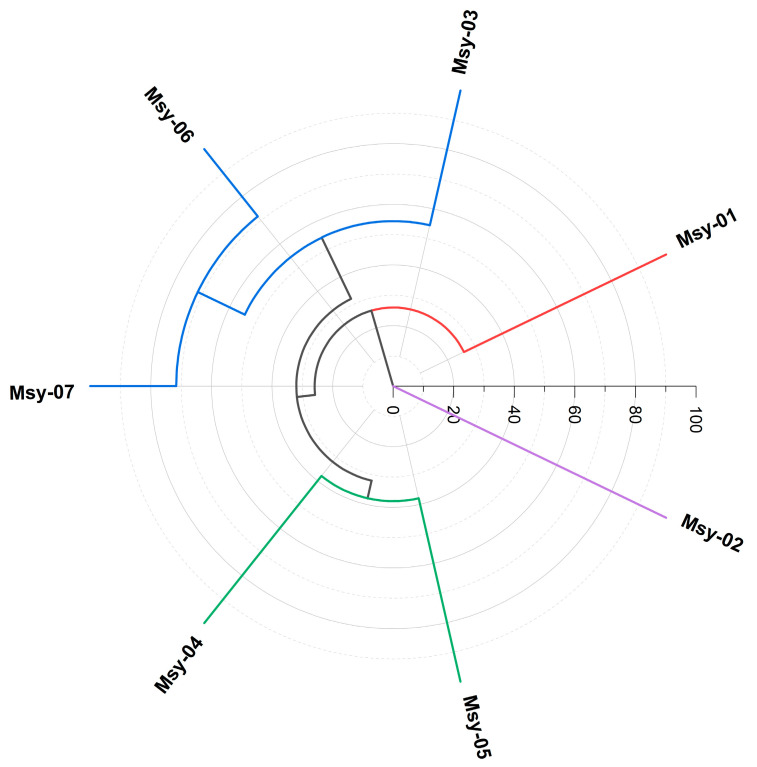
Dendrogram representing the similarity relation of the oil composition of *Myrcia sylvatica*.

**Figure 3 molecules-27-08975-f003:**
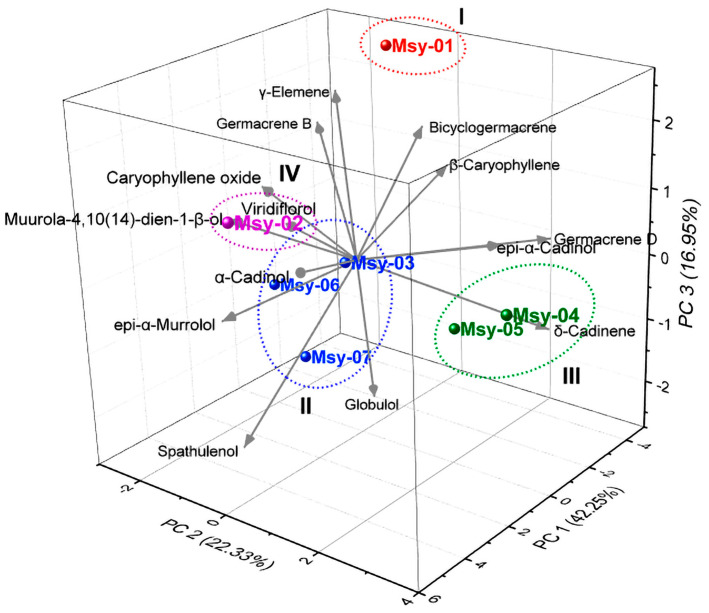
Principal components analysis of the oils of *Myrcia sylvatica*.

**Table 1 molecules-27-08975-t001:** Yield and composition of essential oils from *Myrcia sylvatica* leaves.

	RI_(C)_	RI_(L)_	Constituents (%) *	Msy-01	Msy-02	Msy-03	Msy-04	Msy-05	Msy-06	Msy-07
1	847	846 ^a^	Hex-(*2E*)-enal	0.1		tr	0.1	0.5	tr	tr
2	850	850 ^a^	Hex-(*3Z*)-enol	0.1	tr	0.1	0.4	0.7	0.1	tr
3	858	859 ^a^	Hex-(2*Z*)-enol				0.1	0.2		
4	862	863 ^a^	*n*-Hexanol	0.1		0.1	0.2	0.5	tr	tr
5	904	907 ^a^	Butyl propanoate		0.1					
6	934	932 ^a^	α-Pinene				0.1	1.2	0.1	
7	977	974 ^a^	β-Pinene				0.1	0.4	tr	
8	991	988 ^a^	Myrcene				0.1	0.7	tr	
9	1028	1024 ^a^	Limonene				tr	0.2	0.1	tr
10	1190	1186 ^a^	α-Terpineol				0.1	0.1		
11	1194	1190 ^a^	Methyl salicylate				0.1			
12	1195	1195 ^a^	Myrtenal				0.1			
13	1338	1335 ^a^	δ-Elemene	0.4		0.1	0.3	0.8	0.1	0.1
14	1349	1345 ^a^	α-Cubebene			0.1	0.2	0.4	tr	
15	1352	1350 ^a^	Citronellyl acetate						0.1	
16	1368	1367 ^b^	Cyclosativene			0.1	0.2			0.1
17	1371	1373 ^a^	α-Ylangene			0.1	0.3	0.1	tr	0.1
18	1374	1374 ^a^	Isoledene	tr				0.1		
19	1377	1374 ^a^	α-Copaene	0.1	0.3	0.1	1.8	2.8	0.3	0.2
20	1381	1378 ^a^	Hex-(3*Z*)-enyl hexenoate				0.1			
21	1386	1387 ^a^	β-Bourbonene	0.2	0.1	0.3	0.8	0.6	0.2	0.2
22	1390	1387 ^a^	β-Cubebene				0.1			
23	1392	1389 ^a^	β-Elemene	1.3	0.9	1.1	1.8	1.4	1.6	0.9
24	1410	1409 ^a^	α-Gurjunene	0.1		tr	0.1	0.3	0.1	
**25**	**1422**	**1417 ^a^**	**β-Caryophyllene**	**10.0**	**2.4**	**4.3**	**2.5**	**10.1**	**1.8**	**4.7**
26	1428	1428 ^a^	(*E*)-α-Ionone			0.2	0.2	0.1		
27	1429	1430 ^a^	β-Copaene	0.2	0.1	0.3		0.5	0.1	0.1
28	1429	1431 ^a^	β-Gurjunene				0.9			
**29**	**1434**	**1434 ^a^**	**γ-Elemene**	**12.5**	**0.3**	**2.9**	**0.7**		**7.6**	**2.9**
30	1437	1432 ^a^	α-*trans*-Bergamotene	0.6	0.3	0.3	0.3	1.0		
31	1440	1439 ^a^	Aromadendrene	0.4	0.4	0.2	0.2	0.9	0.4	0.2
32	1440	1437 ^a^	α-Guaiene				0.5			
33	1443	1442 ^a^	Guaia-6,9-diene					0.6		
34	1444	1445 ^b^	Selina-5,11-diene	0.1						
35	1447	1448 ^a^	*cis*-Muurola-3,5-diene	tr						0.5
36	1451	1447 ^a^	Isogermacrene D				0.2	0.2		
37	1451	1451 ^a^	*trans*-Muurola-3,5-diene	0.1				0.5		
38	1454	1452 ^a^	α-Humulene	1.4	0.4	1.1	1.6	1.2	1.9	1.4
39	1462	1460 ^a^	*allo*-Aromadendrene	0.6	0.2					
40	1461	1464 ^a^	9-*epi*-(*E*)-Caryophyllene			0.3	0.9	3.9	0.6	0.3
41	1463	1472 ^b^	*cis*-Cadina-1(6),4-diene				0.2			
42	1463	1465 ^a^	*cis*-Muurola-4(14),5-diene			0.1				
43	1466	1471 ^a^	Dauca-5,8-diene				0.1			
44	1474	1475 ^a^	*trans*-Cadina-1(6),4-diene	0.2			0.2			
45	1477	1478 ^a^	γ-Muurolene	0.6	0.8	0.9	3.0	1.4		0.7
46	1481	1483 ^a^	α-Amorphene		0.2					
47	1482	1484 ^a^	Germacrene D	3.8		3.5	5.0	4.9	0.5	1.1
48	1485	1476 ^b^	Selina-4,11-diene						0.7	
49	1487	1492 ^a^	β-Selinene	0.6	0.8	0.8	2.1	0.4	1.1	0.7
50	1491	1491 ^a^	10,11-epoxy-Calamenene						0.2	0.2
51	1491	1493 ^a^	*trans*-Muurola-4(14),5-diene				0.5			
52	1492	1489 ^a^	δ-Selinene	0.3						
53	1496	1496 ^a^	Viridiflorene	2.0	1.1	1.2	3.1	2.3	2.2	1.2
54	1497	1500 ^a^	Bicyclogermacrene	5.0		0.8		3.8	1.4	0.8
55	1501	1500 ^a^	α-Muurolene	0.5	1.8	0.8	2.7	1.5	0.9	0.5
56	1508	1502 ^a^	*trans*-β-Guaiene					1.0		
57	1508	1509 ^a^	α-Bulnesene				0.4		0.1	
58	1508	1511 ^a^	δ-Amorphene	0.4						0.2
59	1509	1505 ^a^	β-Bisabolene		0.2				0.9	
60	1515	1513 ^a^	γ-Cadinene	0.7	0.7	1.0	2.6	0.9	0.4	0.7
61	1519	1514 ^a^	Cubebol						0.3	0.2
62	1523	1521 ^a^	*trans*-Calamenene		0.6					
**63**	**1524**	**1522 ^a^**	**δ-Cadinene**	**1.8**		**2.7**	**5.6**	**4.8**	**1.1**	**1.3**
64	1531	1532 ^a^	γ-Cuprenene						0.1	
65	1533	1533 ^a^	*trans*-Cadina-1,4-diene	0.1		tr	0.1	tr		
66	1536	1528 ^a^	Zonarene				0.3		0.2	0.2
67	1539	1537 ^a^	α-Cadinene		0.2		0.7	0.3		
68	1539	1540 ^b^	Selina-4(15),7(11)-diene	1.6		0.6			1.5	0.7
69	1543	1545 ^a^	Selina-3,7(11)-diene	1.9	0.2				1.5	0.6
70	1544	1544 ^a^	α-Calacorene			1.4	1.9	0.6	0.5	0.9
**71**	**1558**	**1559 ^a^**	**Germacrene B**	**24.5**	**0.7**	**7.8**	**1.3**	**0.3**	**20.7**	**7.9**
72	1558	1562 ^a^	*epi*-Longipinanol					0.3		
73	1563	1564 ^a^	β-Calacorene			0.3	0.2	0.1		
74	**1568**	**1567 ^a^**	Palustrol	0.8	3.3			1.8		
**75**	**1578**	**1577 ^a^**	**Spathulenol**	**2.9**	**13.4**	**11.1**	**9.8**	**10.1**	**15.7**	**16.0**
**76**	**1584**	**1582 ^a^**	**Caryophyllene oxide**	**3.3**	**15.0**	**9.6**	**3.7**	**0.4**	**0.1**	**0.6**
**77**	**1592**	**1592 ^a^**	**Viridiflorol**	**1.5**	**5.3**	**1.0**	**1.2**	**2.9**	**0.9**	**0.8**
**78**	**1583**	**1590 ^a^**	**Globulol**					**7.4**	**3.4**	**5.5**
79	1595	1595 ^a^	Cubeban-11-ol	0.7	2.9		0.9	1.1	0.6	0.7
80	1602	1600 ^a^	Rosifoliol	0.9	3.0	0.7	0.9	1.5	0.8	0.9
81	1609	1608 ^a^	Humulene epoxide		0.9	1.0	0.5	0.4	0.2	0.3
82	1615	1618 ^a^	1,10-di-*epi*-Cubenol			0.3	0.4	0.4	1.2	1.0
83	1618	1618 ^a^	Junenol		1.3	0.9	1.1	0.6		
84	1629	1627 ^a^	1-*epi*-Cubenol			1.7	2.6	2.6		1.8
**85**	**1629**	**1630 ^a^**	**Muurola-4,10(14)-dien-1β-ol**		**5.8**					
86	1629	1632 ^a^	α-Acorenol	0.8						
87	1633	1630 ^a^	γ-Eudesmol	0.3					0.6	1.0
88	1637	1639 ^a^	Caryophylla-4(12),8(13)-dien-5β-ol			0.7				0.6
89	1643	1645 ^a^	Cubenol						0.3	0.7
90	1643	1640 ^a^	*epi*-α-Murrolol		4.8	3.2		2.2	1.2	2.3
**91**	**1643**	**1640 ^b^**	***epi*-α-Cadinol**	**1.6**	**1.4**		**5.1**	**1.2**		
92	1647	1644 ^a^	α-Muurolol	0.4	2.3	1.1	2.0	1.2		
93	1649	1648 ^a^	*cis*-Guaia-3,9-dien-11-ol		0.9					
**94**	**1655**	**1652 ^a^**	**α-Cadinol**	**1.9**	**8.9**	**3.3**	**5.5**	**2.5**	**1.7**	**3.1**
95	1666	1668 ^b^	Intermedeol	0.9	0.7	0.8			0.8	1.0
96	1668	1668 ^a^	*trans*-Calamenen-10-ol				0.1	0.1		
97	1671	1668 ^a^	14-hydroxy-9-*epi*-(*E*)-Caryophyllene			1.9				
98	1675	1675 ^a^	Cadalene		0.5	0.6	0.8			
99	1677	1676 ^a^	Mustakone				0.7			
100	1685	1679 ^a^	Kusinol				0.7			
101	1686	1664 ^a^	Longiborneol acetate	0.1						
102	1690	1685 ^a^	Germacra-4(15),5,10(14)-trien-1α-ol		0.2		0.1	0.2		
103	1696	1696 ^b^	Juniper camphor	1.5	1.6	2.0	0.2		2.1	3.3
104	1701	1702 ^a^	10-*nor*-Calamenen-10-one				0.2			
105	1739	1733 ^a^	Isobicyclogermacrenal				0.2			0.1
106	1762	1766 ^a^	Drimenol			0.2				
107	1771	1767 ^a^	14-oxy-α-Muurolene				0.2			
108	1780	1779 ^a^	14-hydroxy-α-Muurolene			0.1	0.1	tr		0.2
109	1798	1792 ^a^	β-Eudesmol acetate			0.2				
110	1801	1803 ^a^	14-hydroxy-δ-Cadinene			tr	0.1			0.1
111	1836	1845 ^a^	(2*E*,6*E*)-Farnesyl acetate					0.1		
112	2113	2106 ^b^	Phytol	0.1		0.1	0.3			
	Monoterpene hydrocarbons			-	-	-	0.3	2.5	0.2	tr
	Oxygenated monoterpenoids			-	-	tr	0.3	0.1	0.1	-
	Sesquiterpene hydrocarbons			71.8	12.5	33.1	43.5	47.7	48.0	29.0
	Oxygenated sesquiterpenoids			17.4	71.5	39.7	36.0	36.7	29.9	40.4
	Others			0.4	0.7	1.3	2.5	2.1	0.2	0.2
	Total			89.6	84.7	74.1	82.5	89.2	78.4	69.6
	Oil yield (%) *			0.7	0.9	0.6	0.3	0.3	0.3	0.5

RI_(C)_ = calculated retention index using an *n*-alkane standard solution (C8–C40) in Rtx-5MS column; RI_(L)_ = literature retention index. * Main constituents in bold, n = 2 (standard deviation was less than 2.0% in chemical composition and <0.1% in oil yield); tr = traces (% < 0.1); Msy = *Myrcia sylvatica*; ^a^ = Adams library [15]; ^b^ = FFNCS library [16].

**Table 2 molecules-27-08975-t002:** Collection site, herbarium voucher number, and geographic coordinates for the *Myrcia sylvatica* specimens.

Code	Voucher Number	Coordinates Latitude/Longitude
Msyl-1	MG-228738	1°15′52.65″S, 48°28′12.85″W
Msyl-2	MG-229217	1°14′52.69″S, 48°26′30.20″W
Msyl-3	MG-229955	1°15′52.42″S, 48°28′12.58″W
Msyl-4	MG-229956	1°15′52.41″S, 48°28′12.69″W
Msyl-5	MG-229954	1°15′42.54″S, 48°28′1.78″W
Msyl-6	MG-233283	1°14′51.71″S, 48°26′29.66″W
Msyl-7	MG-233284	1°14′20.79″S, 48°26′9.94″W

## Data Availability

The data presented in this study are available on request from the corresponding author.

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
