# Peer review of "Variability in the Chemical Composition of Myrcia sylvatica (G. Mey) DC. Essential Oils Growing in the Brazilian Amazon"

_molecules, 2022, doi:10.3390/molecules27248975_

Round 1

Reviewer 1 Report

The manuscript entitled "Variability in the Chemical Composition of Myrcia sylvatica (G. Mey) DC. Essential Oils Growing in the Brazilian Amazon" found a chemical variability in the oils of Myrcia sylvatica occurring in the Amazon region. The research results had a certain reference value. However, there are some questions and details that need to be modified:

As far as I know, the extraction methods of essential oil from Myrica sylvatica leaves will affect the concentration and type of essential oil. Please authors explain why to choose the method of extraction process mentioned in the manuscript.

More classification results of Myrica sylvatica have been found by the authors. However, the difference in concentration and species will further cause changes in the functional characteristics of essential oil. If authors could further summarize whether different classifications lead to the variation in functional characteristics of essential oil, the quality of MS will be further improved.

Finally, the author should propose future research directions. How to advance the chemotaxonomy and economical use of Myrica sylvatica.

Author Response

Reviewer 1

The manuscript entitled "Variability in the Chemical Composition of Myrcia sylvatica (G. Mey) DC. Essential Oils Growing in the Brazilian Amazon" found a chemical variability in the oils of Myrcia sylvatica occurring in the Amazon region. The research results had a certain reference value. However, there are some questions and details that need to be modified:

As far as I know, the extraction methods of essential oil from Myrica sylvatica leaves will affect the concentration and type of essential oil. Please authors explain why to choose the method of extraction process mentioned in the manuscript.

R= Dear Reviewer, our objective was to compare the different essential oils profiles of the seven essential oils extracted by hydrodistillation. The literature (five articles, eight chemical profiles) describes only hydrodistillation method to extract the Myrcia sylvatica essential oil. Moreover, our results are much more accurate once the statistical analysis were performed using the data from the seven M. sylvatica samples extracted by hydrodistillation. Furthermore, the objective was to carry out a chemotaxonomic study comparing our result with the samples described in literature, and our methodology agrees with several articles published in the literature as follows:

- doi:10.1016/j.bse.2018.04.017.

-  doi:10.1590/1983-084X/15_006.

- doi:10.1002/ffj.1242.

- doi:10.9734/ejmp/2016/25494.

-  doi:10.3390/antiox11102076.

More classification results of Myrica sylvatica have been found by the authors. However, the difference in concentration and species will further cause changes in the functional characteristics of essential oil. If authors could further summarize whether different classifications lead to the variation in functional characteristics of essential oil, the quality of MS will be further improved.

R= corrected, see line 168-172

Finally, the author should propose future research directions. How to advance the chemotaxonomy and economical use of Myrica sylvatica.

R= We accept the reviewer’s suggestions; we add the propose future research directions in conclusions.

Reviewer 2 Report

This study presents the variability in the chemical composition of essential Oils from 7 specimens of Myrcia sylvatica collected in the Brazilian Amazon. This manuscript is suitable to be published in Molecules, however, before its publication it needs to correct some errors and address some issues as detailed below.

 -Line 20: Information about the species Myrcia sylvatica must be given.

-Line 30: It is suggested not to use apostrophes "study's"

-Line 46: Change "they”

-line 54: Use quotes "kumate-folha-miúda"

-Line 48 and 54: It is suggested to indicate the meaning of the vernacular names

-Line 57-62: This information is from Murcia species, it should be in paragraph 2 or 3.

-The introduction is incomplete, the traditional and industrial uses of the Myrcia sylvatica species should be indicated.

-Is the species wild or cultivated? If it is cultivated, indicate quantities

-What is the status of the species in Brazil, native, endemic, etc.?

-Line 65: It is stated that "in view of the potential presented by Myrcia sylvatica" but this (potential) is not detailed in the introduction

-There is no lower limit in the percentage of identification, however most of the articles present percentages higher than 95%, in this investigation there are percentages of around 70%, which means that 30% of the composition was not identified. It is suggested to improve the percentage of identification.

-Line 77: The postulate "standard deviation was less than 2.0" is not clear

-Deviations in yields should be indicated.

-Explain the contribution of Figure 1 to the investigation.

-What do the colors in figure 1 mean?

Why was Bicyclogermacrene not considered in group 1?

-Chemotypes are samples characterized by different chemical compounds, however β-caryophyllene appears in three different groups. Similar to this case there are others that must be reviewed.

-The definition of the chemotypes is very weak, a better definition must be made, differentiating between groups, profiles and chemotypes.

Author Response

Reviewer 2

This study presents the variability in the chemical composition of essential Oils from 7 specimens of Myrcia sylvatica collected in the Brazilian Amazon. This manuscript is suitable to be published in Molecules, however, before its publication it needs to correct some errors and address some issues as detailed below.

 -Line 20: Information about the species Myrcia sylvatica must be given.

R= We accept the reviewer’s suggestions; corrected

-Line 30: It is suggested not to use apostrophes "study's"

R= corrected

-Line 46: Change "they”

R= corrected

-line 54: Use quotes "kumate-folha-miúda"

R= corrected

-Line 48 and 54: It is suggested to indicate the meaning of the vernacular names

R= The vernacular name probably comes from Tupi Guarani, but we couldn't find the meaning.

-Line 57-62: This information is from Myrcia species, it should be in paragraph 2 or 3.

R= corrected

-The introduction is incomplete, the traditional and industrial uses of the Myrcia sylvatica species should be indicated.

R= Myrcia sylvatica is used Only in traditional medicine, without industrial uses. The traditional uses were described

-Is the species wild or cultivated? If it is cultivated, indicate quantities

R= We collected wild specimens. We insert this information in material and methods, and results sections

-What is the status of the species in Brazil, native, endemic, etc.?

R= Corrected, see line 61

-Line 65: It is stated that "in view of the potential presented by Myrcia sylvatica" but this (potential) is not detailed in the introduction

R= Corrected, we talk about biological potential, see previous paragraph

-There is no lower limit in the percentage of identification, however most of the articles present percentages higher than 95%, in this investigation there are percentages of around 70%, which means that 30% of the composition was not identified. It is suggested to improve the percentage of identification.

R= Myrcia sylvatica was previously investigated by our research group (see reference 12), and the plant has a complex essential oil composition, with a mixture of many sesquiterpenes. We could not improve the percentage of identification.

-Line 77: The postulate "standard deviation was less than 2.0" is not clear.

R= Our experimental design used analytical duplicates, that is, a single specimen monthly collected, extracted, and analyzed in duplicate. The table 1 shows the mean, the standard deviation was less than 2.0, and was therefore not included in the table.

-Deviations in yields should be indicated.

R= Corrected, see the table 1 footnote.

-Explain the contribution of Figure 1 to the investigation.

R= Figure 1 shows the biosynthetic pathway of the main constituents identified if the samples. The elucidation of biosynthesis is fundamental for understanding the diversity of terpenes that a cation can generate, allowing the understanding of the terpene skeletons present in specimens of M. sylvatica.

-What do the colors in figure 1 mean?

R= Each color represents a cation that originates each constituent.

-Why was Bicyclogermacrene not considered in group 1?

- Although the Bicyclogermacrene was identified in the group I sample, based on the statistical analysis, it is only weakly related to this group.

-Chemotypes are samples characterized by different chemical compounds, however β-caryophyllene appears in three different groups. Similar to this case there are others that must be reviewed.

R= Chemometric analyzes such as Principal Component Analysis (PCA) and Hierarchical Cluster Analysis (HCA) are techniques used to group essential oils from multiple samples. Therefore, based on the statistical analysis, the chemotypes were distinguished from each other. Although β-caryophyllene was identified in all samples, using the selected compounds (>4.0%), it was strongly related only in groups I and III.

-The definition of the chemotypes is very weak, a better definition must be made, differentiating between groups, profiles and chemotypes.

R= The samples were grouped based on multivariate statistical analyses, including Principal Component Analysis (PCA) and Hierarchical Cluster Analysis (HCA). Well-established and implemented methods, used by other recently published studies:

- https://doi.org/10.1016/j.indcrop.2020.112449

- https://doi.org/10.1016/j.bse.2022.104543

- https://doi.org/10.1016/j.jksus.2021.101634

- https://doi.org/10.1016/j.bse.2021.104323

Reviewer 3 Report

The manuscript "Variability in the Chemical Composition of Myrcia sylvatica (G. Mey) DC. Essential Oils Growing in the Brazilian Amazon" is very interesting and readible.

In my opininon, only few small comments:

Abstract: Instead of "Myrcia species have great ecological, economic, food, and medicinal relevance" start with "Myrcia sylvatica..."

Material and method: It is unclear whether the plants are wild grown or cultivated. Clarify.

In addition, in conclusion, state whether this plant is cultivated, on which areas, perspectives for the future, i.e. which of the detected hemotypes has the greatest potential for application.

Author Response

Reviewer 3

The manuscript "Variability in the Chemical Composition of Myrcia sylvatica (G. Mey) DC. Essential Oils Growing in the Brazilian Amazon" is very interesting and readible.

In my opininon, only few small comments:

Abstract: Instead of "Myrcia species have great ecological, economic, food, and medicinal relevance" start with "Myrcia sylvatica..."

R= We accept the reviewer’s suggestion.

Material and method: It is unclear whether the plants are wild grown or cultivated. Clarify.

R= Corrected,

In addition, in conclusion, state whether this plant is cultivated, on which areas, perspectives for the future, i.e. which of the detected hemotypes has the greatest potential for application.

R= Corrected,

Round 2

Reviewer 2 Report

The manuscript has been improved, however there are still points that must be addressed

-Line 78, what are the units of 2 and 0.1?

-Line 89, Figure 1?

-The grouping of samples based on multivariate statistical analyses, including Principal Component Analysis (PCA) and Hierarchical Cluster Analysis (HCA) is what is expected in this type of research. The question is, what is the difference between groups, profiles and chemotypes, the authors use these three terms indiscriminately, are they the same?

Author Response

The manuscript has been improved, however there are still points that must be addressed

R: Dear reviewer, thanks for your contributions

-Line 78, what are the units of 2 and 0.1?

R: The unit of measure of the standard deviation is equal to the unit of measure of the observations in the sample. Therefore, analytically the standard deviation unit is % area for chemical composition, and % (v/w) for oil yield.

-Line 89, Figure 1?

R: Thanks a lot for your observations. Corrected.

-The grouping of samples based on multivariate statistical analyses, including Principal Component Analysis (PCA) and Hierarchical Cluster Analysis (HCA) is what is expected in this type of research. The question is, what is the difference between groups, profiles and chemotypes, the authors use these three terms indiscriminately, are they the same?

R: The terms "Profiles" or "chemotypes" are used to classify the volatiles samples. Although "chemotype" term is used when chemometrics are applied. The "Group" (cluster) is the classification or grouping obtained in multivariate analysis. Since we compared our samples with samples from the literature, we thought it prudent to make these distinctions.